# NORML: Nodal Optimization for Recurrent Meta-Learning

## Abstract

Meta-learning is an exciting and powerful paradigm that aims to improve the effectiveness of current learning systems. By formulating the learning process as an optimization problem, a model can learn how to learn while requiring significantly less data or experience than traditional approaches. Gradient-based meta-learning methods aims to do just that, however recent work have shown that the effectiveness of these approaches are primarily due to *feature reuse* and very little has to do with priming the system for *rapid learning* (learning to make effective weight updates on unseen data distributions). This work introduces Nodal Optimization for Recurrent Meta-Learning (NORML), a novel meta-learning framework where an LSTM-based *meta-learner* performs neuron-wise optimization on a *learner* for efficient task learning. Crucially, the number of *meta-learner* parameters needed in NORML, increases linearly relative to the number of *learner* parameters. Allowing NORML to potentially scale to *learner* networks with very large numbers of parameters. While NORML also benefits from *feature reuse* it is shown experimentally that the meta-learner LSTM learns to make effective weight updates using information from previous data-points and update steps.

## 1 Introduction

Humans have a remarkable capability to learn useful concepts from a small number of examples or a limited amount of experience. In contrast most machine learning methods require large, labelled datasets to learn effectively. Little is understood about the actual learning algorithm(s) used by the human brain, and how it relates to machine learning algorithms like backpropagation (Lillicrap & Körding (2019)). Botvinick et al. (2019) argues that inductive bias and structured priors are some of the main factors that enable fast learning in animals. In order to build general-purpose systems we must be able to design and build learning algorithms that can quickly and effectively learn from a limited amount of data by utilizing prior knowledge and experience.

Supervised *few-shot learning* aims to challenge machine learning models to learn new tasks by leveraging only a handful of labelled examples. Vinyals et al. (2016) introduces the few-shot learning problem for image classification, where a model is tasked to classify a number of images while being provided either 1 or 5 examples of each class (hereafter referred to as 1-shot and 5-shot learning). One way to approach this problem is by way of *meta-learning*, a broad family of techniques that aim to learn how to learn (Thrun & Pratt (1998)). One particularly powerful group of approaches, known as memory-based methods, use memory architectures that can leverage prior information to assist in learning (Santoro et al. (2016); Ravi & Larochelle (2017)). Optimization-based methods (Finn et al. (2017)) is another exciting area that aim to learn an initial set of parameters that can quickly adapt to new, unseen tasks with relatively little training data.

This work introduces a novel technique where a recurrent neural network based *meta-learner* learns how to make parameter updates for a task *learner*. The *meta-learner* and the *learner* are jointly trained so as to learn how to learn new tasks with little data. This approach allows one to utilize aspects from both optimization-based and memory-based meta-learning methods. The recurrent architecture of the meta-learner can use important prior information when updating the *learner*, while the *learner* can learn a set of initial task parameters are easily optimized for a new task. The vanishing gradient challenge faced by gradient-based optimization is solved by using a Long short-term memory (LSTM) based *meta-learner* (Hochreiter & Schmidhuber (1997)). Memory-based

methods (Ravi & Larochelle (2017)) that use all of the *learner*'s parameters as input to a *meta-learner* tend to break down when using a *learner* with a large number of parameters (Andrychowicz et al., 2016). The technique proposed in this work, Nodal Optimization for Recurrent Meta-Learning (NORML), solves this scaling problem by learning layer-wise update signals used to update the *learners*'s parameters.

NORML is evaluated on the Mini-ImageNet dataset and is shown to improve on existing optimization-based and memory-based methods. An ablation study is done showing the effects of the different components of NORML. Furthermore a comparison is done between NORML and Model Agnostic Meta-Learning (MAML) using the Omniglot dataset. The comparison demonstrates that NORML makes superior parameter updates than the updates made by gradient descent.

## 2 PRELIMINARIES

### 2.1 BACKPROPAGATION

The backpropagation algorithm (Rumelhart et al., 1986) was developed in the 1980's and has since been the status quo for training neural networks. By recursively applying the chain rule, backprop-agation sends gradient signals back through stacked layers of a deep network. Each gradient signal at a particular layer is used to calculate the gradient of the weights connected to that layer.

Consider a neural network with N hidden layers. Let the rows of matrix $\boldsymbol{W}_l$ denote the weights connecting the nodes in the layer below to a hidden node in layer $l$ and let the vector $\boldsymbol{b}_l$ denote the biases connected to each node in layer $l$. Then:

$$\boldsymbol{a}_l = \boldsymbol{W}_l \boldsymbol{h}_{l-1} + \boldsymbol{b}_l, \boldsymbol{h}_l = f_l(\boldsymbol{a}_l) \tag{1}$$

Where $f_l$ is the activation function used at layer $l$, $\boldsymbol{a}_l$ is the layer's pre-activation output, and $\boldsymbol{h}_l$ is referred to as the *activations* of layer $l$. Note that $\boldsymbol{h}_0 = \boldsymbol{x}$ is the input to the network. The output of the network is given by $\hat{\boldsymbol{y}} = f_{out}(\boldsymbol{a}_{out})$. When choosing to use a softmax as the output activation and a cross-entropy loss function, the loss and the output layer's gradient are calculated as follows:

$$L = -\frac{1}{n} \sum_n \boldsymbol{y}_n \log \hat{\boldsymbol{y}}_n + (1 - \boldsymbol{y}_n) \log(1 - \hat{\boldsymbol{y}}_n) \tag{2}$$

$$\boldsymbol{\delta}_{\boldsymbol{a}_{out}} = \frac{\partial L}{\partial \boldsymbol{a}_{out}} = \hat{\boldsymbol{y}} - \boldsymbol{y} \tag{3}$$

where $L$ is the cross-entropy loss, $n$ is the network's output size, and $\boldsymbol{\delta}_{\boldsymbol{a}_{out}}$ is the gradient at the pre-activation of the output layer. Note that $\boldsymbol{\delta}_{\boldsymbol{a}_{out}}$ is an n-dimensional vector where each element of $\boldsymbol{\delta}_{\boldsymbol{a}_{out}}$ describes the corresponding node's contribution to the loss. By propagating $\boldsymbol{\delta}_{\boldsymbol{a}_{out}}$ back through the network one obtains the gradients at each of the hidden layers' nodes:

$$\boldsymbol{\delta}_{\boldsymbol{a}_{l-1}} = \frac{\partial L}{\partial \boldsymbol{a}_{l-1}} = (\boldsymbol{W}_l^T \boldsymbol{\delta}_{\boldsymbol{a}_l}) \odot f'_{l-1}(\boldsymbol{a}_{l-1}) \tag{4}$$

where $f'_l()$ is the derivative of the activation function, $\odot$ is an element-wise multiplication and $\boldsymbol{\delta}_{\boldsymbol{a}_l}$ is the pre-activation gradient at layer $l$. In order to calculate the gradient with regards to the weights, $\boldsymbol{\delta}_{\boldsymbol{a}_l}$ is matrix multiplied with the activations of the previous layer:

$$\frac{\partial L}{\partial \boldsymbol{W}_l} = \boldsymbol{\delta}_{\boldsymbol{a}_l} \boldsymbol{h}_{l-1} \tag{5}$$

### 2.2 META-LEARNING PROBLEM DEFINITION

The $N$-way $K$-shot problem is defined using an episodic formulation as proposed in Vinyals et al. (2016). A task $T_i$ is sampled from a task distribution $p(T)$ and consists of an $N$-way classification problem. A meta-dataset is divided into a *training meta-set* $S^{tr}$, a *validation meta-set* $S^{val}$ and a *testing meta-set* $S^{test}$ where the classes contained in each meta-set is disjointed (i.e. none of the classes in $S^{test}$ is present in $S^{tr}$ and visa versa).

The task $T_i$ consists of a training set $D^{tr}$ and validation set $D^{val}$. The training set $D^{tr}$ contains $K$ examples from each of $T_i$'s $N$ classes. The validation set $D^{val}$ usually contains a larger set of examples from each class in order to give an estimate of the model's generalization performance on task $T_i$. Note that a task's validation set $D^{val}$ (used to optimize the meta training objective) should not be confused with the meta validation set $S^{val}$, which is used for model selection.

## 2.3 MODEL AGNOSTIC META-LEARNING

Optimization-based meta-learning is an approach to meta-learning where an inner loop is used for fast adaptation to a new task, and an outer loop is used to optimize the inner loop's training steps. Model Agnostic Meta-Learning (MAML) is one such approach that contains all the key ingredients used in optimization-based meta-learning. The inner loop consists of training a model, $f_\theta$ via gradient descent on a few-shot learning task, $T_i$.

$$\theta' = G(\theta, D^{tr}) \tag{6}$$

where $G$ is often implemented as a single step of gradient descent on the training set $D^{tr}$; $\theta'_i = \theta - \eta \nabla_\theta L^{tr}_{T_i}(f_\theta)$. It is often advantageous for $G$ to consist of multiple sequential update steps.

The validation set of $T_i$ is then used to evaluate $f_{\theta'_i}$, and the task-specific validation loss, $L^{val}_{T_i}$, is calculated. The outer loop consists of optimizing the base-parameters $\theta$, using the sum of $M$ different tasks' validation losses:

$$\theta \leftarrow \theta - \eta \nabla_\theta \sum_{T_i \sim p(T)}^{M} L^{val}_{T_i}(f_{\theta'_i}) \tag{7}$$

where $M$ is referred to as the meta-batch size.

This approach allows a model to learn a set of base-parameters $\theta$, that can quickly adapt to a new unseen task. After meta-training a model, the inner loop procedure can update $\theta$ and return task-specific parameters $\theta'_i$ using only a few examples of the new task. By differentiating through the inner loop, MAML learns to update the base-parameters $\theta$ in such a way that the task-specific parameters, $\theta'_i$, generalizes to unseen examples of task $T_i$.

In many cases it is preferred to have an inner loop that consists of multiple sequential updates. However the inner loop's computational graph can become quite large if too many steps are taken. This often results in exploding and vanishing gradients since the outer loop still needs to differentiate through the entire inner loop Aravind Rajeswaran (2019) Antoniou et al. (2018). This limits MAML to domains where a small amount of update steps are sufficient for learning. The LSTM-based *meta-learner* proposed in this work, allow gradients to effectively flow through a large number of update steps. NORML can therefore be applied to a wide array of domains.

## 3 MODEL

### 3.1 TRAINING

NORML incoporates a neural network based *learner* with $N$ layers and an LSTM-based *meta-learner* that is used to optimize the *learner*'s inner loop parameter updates. Figure 1 depicts the process by which the *meta-learner* calculates the parameter updates for a single layer *learner*. When multiple layers are optimized the additional gradient signals and layer inputs are added to the *meta learner*'s input. The *meta-learner* outputs two update signals for every layer, so when a 4-layer network is used as the *learner*, the *meta-learner* outputs 8 update signals.

The high-level operation is as follows (Algorithm 1). First the *learner*'s task-specific loss, $L^{tr}_{T_i}(f_\theta)$, and layer-gradients, $\delta_{a_l}$, are calculated via cross-entropy and backpropagation. The meta-learner takes $\delta_{a_l}$, $h_{l-1}$ and the loss, $L$, as input, and outputs the current cell state and the update signals, $\hat{h}_{l-1}$ and $\hat{\delta}_{a_l}$:

$$\begin{bmatrix} \hat{\delta}_{a_l} \\ \hat{h}_{l-1} \\ c_{l,t+1} \end{bmatrix} = m_\Phi(\delta_{a_l}, h_{l-1}, L, c_{l,t}) \tag{8}$$

The update signals are matrix multiplied to determine the parameter updates for layer $l$:

$$\Delta \boldsymbol{W}_l = \eta \hat{\boldsymbol{\delta}}_{\boldsymbol{a}_l} \hat{\boldsymbol{h}}_{l-1} \tag{9}$$

$$\Delta \boldsymbol{b}_l = \eta \hat{\boldsymbol{\delta}}_{\boldsymbol{a}_l} \tag{10}$$

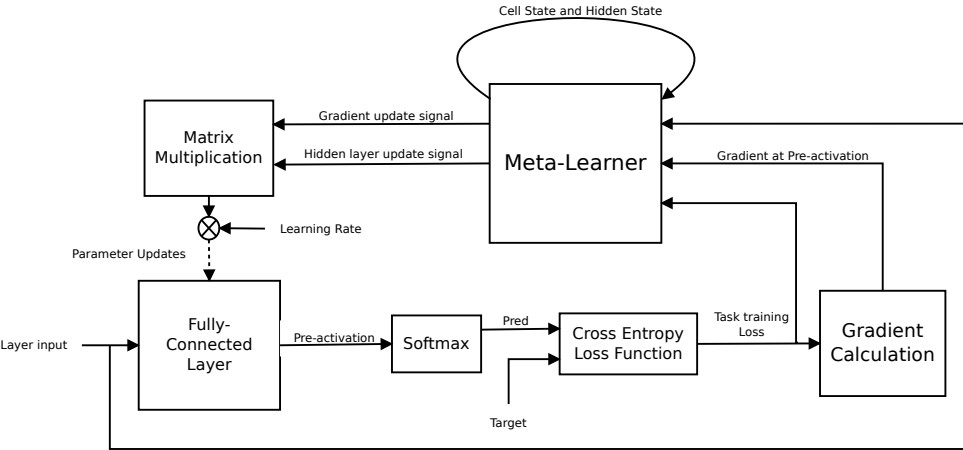

Figure 1: Training process of NORML.

The *learner*'s updated parameters, $\theta'_{T_i}$, are then used on the next training example and the whole process gets repeated for $U$ update steps. The inner loop terminates after calculating the task validation loss using $f_{\theta'}$.

$$L_{T_i}^{val} = CrossEntropy(\boldsymbol{y}_{val}, f_{\theta'}(\boldsymbol{x}_{val})) \tag{11}$$

The outer loop consists of optimizing both $\theta_0$ and the meta-learner parameters $\Phi$ via gradient descent:

$$\theta \leftarrow \theta - \beta \nabla_\theta \sum_{T_i \ p(T)} L_{T_i}^{val}(f_{\theta'_i}) \tag{12}$$

$$\Phi \leftarrow \Phi - \beta \nabla_\Phi \sum_{T_i \ p(T)} L_{T_i}^{val}(f_{\theta'_i}) \tag{13}$$

### 3.2 META-LEARNER

The *meta-learner*, $m$ is implemented using a modified LSTM cell as shown in Figure 2.

The input to the *meta-learner* is normalized, flattened and concatenated. $x_l$ is used to denote concatenating the loss, the layer gradients, and the activations of the previous layer.

The *meta-learner* consists of a forget gate, $\boldsymbol{f}_l$, an input gate, $\boldsymbol{i}_l$ and an update gate, $\tilde{\boldsymbol{c}}_l$. The cell state at update step $t$ is denoted via $\boldsymbol{c}_{l,t}$, and is calculated as follows:

$$\boldsymbol{f}_{l,t} = \sigma(\boldsymbol{W}_{l,f}\boldsymbol{x}_{l,t} + \boldsymbol{b}_{l,f}) \tag{14}$$

$$\boldsymbol{i}_{l,t} = \sigma(\boldsymbol{W}_{l,i}\boldsymbol{x}_{l,t} + \boldsymbol{b}_{l,i}) \tag{15}$$

$$\tilde{\boldsymbol{c}}_{l,t} = tanh(\boldsymbol{W}_{l,\tilde{c}}\boldsymbol{x}_{l,t} + \boldsymbol{b}_{i,\tilde{c}}) \tag{16}$$

$$\boldsymbol{c}_{l,t} = \boldsymbol{f}_{l,t} \odot \boldsymbol{c}_{l,t-1} + \boldsymbol{i}_{l,t} \odot \tilde{\boldsymbol{c}}_{l,t} \tag{17}$$

In order to determine the update signals, the cell state, $\boldsymbol{c}_{l,t}$, is used as input to two separate fully-connected layers followed by a sigmoid activation function and a pointwise multiplication with the original layer gradient and the previous layer's activations:

$$\hat{\boldsymbol{\delta}}_{\boldsymbol{a}_l} = \boldsymbol{\delta}_{\boldsymbol{a}_l} \odot \sigma(\boldsymbol{W}_{l,g}\boldsymbol{c}_{l,t} + \boldsymbol{b}_{l,g}) \tag{18}$$

$$\hat{\boldsymbol{h}}_l = \boldsymbol{h}_l \odot \sigma(\boldsymbol{W}_{l,h}\boldsymbol{c}_{l,t} + \boldsymbol{b}_{l,h}) \tag{19}$$

---

**Algorithm 1** Nodal Optimization for Rapid Meta-Learning

---

**Require:** : Learner $f$ with parameters $\theta$
**Require:** : Meta-learner $m$ with parameters $\Phi$
**Require:** : $p(T)$: distribution over tasks
**Require:** : $\eta, \beta$: step size hyperparameters
1: randomly initialize $\theta, \Phi$
2: **while** not converged **do**
3:     Sample batch of tasks $T_i \sim p(T)$
4:
5:     **for all** $T_i$ **do**
6:         Let $(D_i^{tr}, D_i^{val}) = T_i$
7:         Initialize $\theta'_0 = \theta$
8:
9:         **for** $t = 1, len(D_i^{tr})$ **do**
10:             Compute learner's training loss $L_{T_i}^{tr}(f_{\theta'_{t-1}})$ and layer gradients $\frac{\partial L_{T_i}^{tr}}{\partial a_l}$ for all layers $l$
11:             Compute the update signals: $\hat{h}_{l-1}, \hat{g}_l = m_\Phi(L_{T_i}^{tr}(f_{\theta'_{t-1}}), \frac{\partial L_{T_i}^{tr}}{\partial a_l}, h_{l-1})$
12:             Update task parameters: $\theta'_t \leftarrow \theta'_{t-1} - \eta \hat{g}_l \hat{h}_{l-1}$
13:         **end for**
14:         Compute validation loss $L_{T_i}^{val}(f_{\theta'_t})$
15:     **end for**
16:     Update $\theta \leftarrow \theta - \beta \nabla_\theta \sum_{T_i \ p(T)} L_{T_i}^{val}(f_{\theta'_t})$
17:     Update $\Phi \leftarrow \Phi - \beta \nabla_\Phi \sum_{T_i \ p(T)} L_{T_i}^{val}(f_{\theta'_t})$
18: **end while**

---

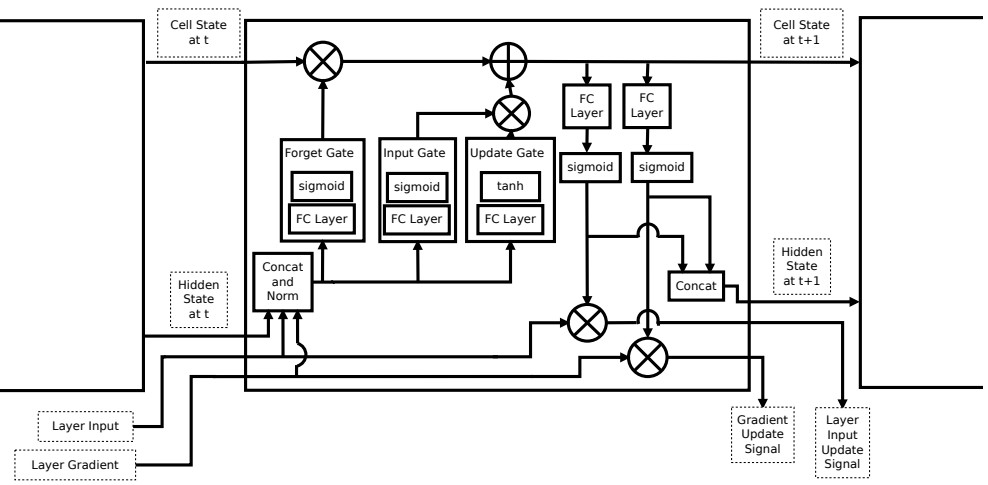

Figure 2: Meta-learner LSTM.

The sigmoid activation and pointwise mutliplication operations allow the *meta-learner* to scale the update signal received by each node in the network. This node-wise scaling let the *meta-learner* control how the weights connected to a particular node will get changed at any given update step. By scaling the layer gradients, the *meta-learner* can dynamically control by how much each node's *input* should be changed at each update step. Likewise, by scaling the hidden layer activations, the *meta-learner* can control the change in weighting of each of the previous nodes' *output*. By controlling the magnitude of both the activations and the layer gradients, the *meta-learner* can learn a *dynamically adaptive learning rate* for each individual weight and bias of the learner. This can be achieved with ***a meta-learner that scales linearly in size relative to the size of the learner***, i.e. increasing the number of *learner* parameters by a factor of $p$, will result in the same factor increase in the number of *meta-learner* parameters.

## 4    RELATED WORK

Learning how to learn has a long history and variety of approaches have been proposed. Memory-based methods use memory architectures to either store key training examples or to encode fast adaptation algorithms (Santoro et al., 2016) (Andrychowicz et al., 2016) (Ravi & Larochelle, 2017). Metric-based methods are designed to learn similarity metrics for examples of the same class (Koch, 2015) (Vinyals et al., 2016) (Snell et al., 2017). Optimization-based methods aim to learn a set of initial parameters that can quickly adapt to new tasks (Finn et al., 2017).

A subset of memory-based methods use recurrent neural networks to learn fast adaptation algorithms. In Ravi & Larochelle (2017) an LSTM-based meta-learner is trained to iteratively optimize a neural network. At a given timestep the meta-learner's hidden state consists of the optimizee's current parameters. At each timestep a training datapoint is used to calculate the optimizee's parameter gradients. The parameter gradients and the task loss is used as input to the meta-learner. The hidden state of the following timestep is then used as the updated parameters for the *learner*. A challenge of this approach is that if you want to optimize tens of thousands of parameters, you would have a massive hidden state and input size, and will therefore require an enormous number of parameters for the meta-learner.

In Andrychowicz et al. (2016) an alternative approach is introduced that avoids the aforementioned scaling problem by individually optimizing each parameter using a separate LSTM. This vastly decreases the number of meta-parameters required, however each meta-learner is only given information of a single parameter and can therefore not use information of other parts of the network when doing optimization. The approach introduced in this work uses a single *meta-learner* to optimize all the parameter's of the *learner*, while the *meta-learner* require less than 5 times the number of parameters used by the *learner*.

Optimization-based methods (Finn et al., 2017), differentiate through the inner loop in order to optimize learning. Improvements to MAML include using a pre-trained network to extract features and to then only optimize the final fully-connected layer via gradient based optimization (**?**). Using a pre-trained feature extractor assists in overcoming task overfitting and have been demonstrated to give a large boost in performance(Sun et al., 2018). In this work a pre-trained feature extractor was also used when evaluating NORML on Mini-Imagenet, but is not a requirement for implementing NORML.

Other work that use custom layer-wise update signals include Nø kland (2016) and Nkland & Eidnes (2019). In feedback alignment layer gradient approximating update signals are used to train a neural network by passing the error signal through fixed random matrices. Backpropagation in contrast passes the error signal through the transpose of the subsequent layer's weight matrix. Surprisingly feedback alignment demonstrates that learning can still occur even when using random matrices for the backward pass.

## 5    EXPERIMENTAL RESULTS

The proposed approach is evaluated on the Mini-Imagenet benchmark dataset. An ablation study comparing the performance against MAML is conducted on the Omniglot dataset. The evaluation aims to compare the performance of NORML against existing methods and to specifically test if node-wise optimization improves the update signals provided by backpropagation.

### 5.1    MINI-IMAGENET CLASSIFICATION

The Mini-Imagenet dataset was proposed by Vinyals et al. (2016) and is a popular benchmark for few-shot learning methods. The dataset consists of 100 classes, each containing 600 images. The meta-training, meta-validation and meta-testing sets each contain 64, 16 and 20 classes respectively. It has been demonstrated that by using a pre-trained feature extractor the performance on few-shot learning tasks can be significantly improved (Sun et al. (2018)). For evaluation of NORML on the Mini-Imagenet benchmark, a Resnet-12 network is pre-trained using only data and classes from the meta-training set. The pre-train model is trained to perform 64-way classification. After pre-training, the final FC-layer is replaced by a randomly initialized FC-layer and the convolutional layers are frozen.

The *meta-learner* network and the final layer of the *learner* are optimized during meta-training. The inner loop consisted of 10 epochs for both the 5-shot and 1-shot cases, while using a meta-batch size of 4. The model was trained for 10 000 meta-iterations and the meta-validation set was used for model selection for testing.

Classification accuracies for NORML and other baselines are shown in Table 1.

| Model | Mini-Imagenet Test Accuracy | |
| --- | --- | --- |
| | 1-shot | 5-shot |
| Matching network (Vinyals et al. (2016)) | 43.56 ±0.84% | 55.31 ±0.73% |
| Meta-learner LSTM (Ravi & Larochelle (2017)) | 43.44 ±0.77% | 60.60 ±0.71% |
| MAML (Finn et al. (2017)) | 48.70 ±1.84% | 63.11 ±0.92% |
| LLAMA (Grant et al. (2018)) | 49.40 ±1.83% | - |
| REPTILE (Nichol et al. (2018)) | 49.97 ±0.32% | 65.99 ±0.58% |
| PLATIPUS (Finn et al. (2018)) | 50.13 ±1.86% | - |
| SNAIL (Mishra et al. (2017)) | 55.71 ±0.99% | 68.88 ±0.92% |
| (Gidaris & Komodakis (2018)) | 56.20 ±0.86% | 73.00 ±0.64% |
| (Bauer et al. (2018)) | 56.30 ±0.40% | 73.90 ±0.30% |
| (Munkhdalai et al. (2017)) | 57.10 ±0.70% | 70.04 ±0.63% |
| (Zhou et al. (2018)) | 58.49 ±0.91% | 71.28 ±0.69% |
| (Qiao et al. (2017)) | 59.60 ±0.41% | 73.74 ±0.19% |
| **NORML** | $\mathbf{60.2 \pm 0.91\%}$ | $\mathbf{74.1 \pm 0.71\%}$ |

Table 1: Test accuracies on the Mini-ImageNet benchmark. The first set of results use convolutional networks, while the second use much deeper residual networks, predominantly in conjunction with pre-training.

## 5.2 ABLATION STUDY

For the Mini-Imagenet benchmark the *learner* consisted of a single FC-layer which was proceeded by a pre-trained feature extractor. NORML can however be applied to networks with multiple layers. To assess the effect of the different components of the *meta-learner* and to compare NORML to MAML, an ablation study was conducted on the Omniglot 20-way dataset. The *learner* consisted of a fully-connected network with 4 hidden layers, each including batch normalization and ReLU activations, followed by a linear layer and softmax. The hidden layer sizes were 256, 128, 64 and 64. The same *learner* network was used for all approaches while only changing the setup of the meta-learner between experiments.

The cell state and hidden state of the *meta-learner* carries information of all the previous losses, gradients and update steps that was calculated for the task at hand. This enables NORML to potentially improve on the update signals calculated via backpropagation, since the gradient for a particular data-point only contains information from the current time step. To assess the contribution of the cell state and hidden state to the performance of NORML, the following three setups were tested: (1) Setting the cell input state and hidden input state to zero, (2) only setting the hidden state to zero and (3) only setting the cell state to zero.

The *meta-learner* used in NORML can learn to dynamically control the learning rates of each *learner* node's input and output weights. To assess the effect of the node-wise learning rates and to compare NORML to MAML, three different training setups are used for MAML: (1) MAML with a constant learning rate, (2) MAML with a single learnable learning rate and (3) MAML with learnable per-layer learning rates. The results for the 1-shot and 5-shot tasks are shown in Table 2. Note that the results of the ablation study are obtained using a *fully-connected network*; a convolutional neural network will give higher accuracies.

| Model | Omniglot 20-way Test Accuracy | |
|---|---|---|
| | 1-shot | 5-shot |
| MAML (constant lr) | 86.3% | 95.7% |
| MAML (global lr) | 86.5% | 95.7% |
| MAML (layer-wise lr) | 87.1% | 96.0% |
| NORML (zero hidden & zero cell) | 86.3% | 95.7% |
| NORML (zero cell) | 86.3% | 95.7% |
| NORML (zero hidden) | 87.9% | 96.5% |
| **NORML** | **90.3**% | **97.2**% |

Table 2: Ablation study and comparison to MAML using different training setups

NORML learns to dynamically control the learning rates of each *learner* node's input and output weights. This is shown to give a significant improvement when compared to using constant, learnable, and per-layer learnable learning rates. Intuitively it makes sense, since NORML can effectively control the learning rates of each parameter in the *learner* network. Additionally NORML can control the learning rates dynamically. Dynamic control seems crucial to NORML's success, since a parameter that receives a large update at step $t$, might need to make a small update at step $t + 1$. The importance of being able to dynamically adapt the learning rate using prior update information, is confirmed in the experiments where the input hidden state and input cell state of the *meta-learner* are set to zero. Although the *meta-learner* can still learn nodal learning rates, it is unable to use information of previous update steps and ends up performing the same as MAML with a constant learning rate.

## 6    CONCLUSION AND FUTURE WORK

Recent work has shown that the successes of most meta-learning techniques are primarily due to *feature reuse* (Raghu et al., 2019). Ideally meta-learning techniques should be learning to also make efficient and effective parameter updates. This work introduces a novel meta-learning method that allows node-wise scaling of *learner* update signals, thereby enabling the meta-learner to dynamically control the learning rate of each of the *learner*'s parameters. Importantly this is done by a *meta-learner* who's size scales linearly relative to the size of the *learner*.

Future work would focus on scaling NORML to optimize large *learner* networks. Furthermore, applying NORML to domains where a large number of update steps are required, such as Deep Reinforcement Learning, could prove to be an exciting avenue of further research since the LSTM-based *meta-learner* allow gradients to propagate back much farther than gradient-based optimization methods (Finn et al., 2017).

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

## A APPENDIX

You may include other additional sections here.

