# OpenReview forum: "NORML: Nodal Optimization for Recurrent Meta-Learning"
_ICLR.cc/2020/Conference — Reject_

### Official Review · AnonReviewer1 · 2019-10-20
**Official Blind Review #1**

**Rating:** 1

**Review:**

This submission proposes NORML, a meta-learning method that 1) learns initial parameters for a base model that leads to good few-shot learning performance and 2) where a recurrent neural network (LSTM) is used to control the learning updates on a small support set for a given task. The method is derived specifically for full connected neural networks, where the meta-learner produces gating factors on the normal gradients (one for each neuron that the parameter is connecting). The method is compared with various published few-shot learning methods on miniImageNet, and an ablation study and detailed comparison with MAML is presented on Omniglot.

Unfortunately, I believe this submission should be rejected, for the following reasons:

1. Limited novelty: it is a fairly incremental variation compared to the Meta-Learner LSTM (Ravi & Larochelle). I actually kind of like the proposed variant, but then I would at least expect a direct comparison with Meta-Learner LSTM. And though the authors try to make a case for why NORML is better than Meta-Learner LSTM, unfortunately they don't actually provide a fair comparison. Indeed, the results for Meta-Learner LSTM on miniImageNet use a very different base learner (i.e. not a ResNet-12) which wasn't pretrained. Same in fact can be said about many of the results reported in Table 1.

2. Missing baselines: though the comparison in Table 2 with MAML is a good step, I actually believe the 3 versions of MAML considered aren't appropriate. Instead, I believe that 1) at a minimum, a comparison with a version of MAML where a separate learning rate is learned "per hidden unit" AND "per inner loop step" is necessary, as it is closer to what NORML can achieve, and 2) a comparison with MAML++ (Antoniou et al.) would be ideal. Finally, I think this study should also be done on miniImageNet, not only Omniglot.

3. Limited applicability: NORML assumes that the base learner is a fully connected network. That strongly limits the applicability of the method, given that most few-shot learners usually have convolutional layers.

4. Inaccurate descriptions of some prior work: The related work includes certain statements that I believe aren't accurate. For example, it is stated that the Meta-Learner LSTM requires "an enormous number of parameters". I believe this is not true: there is a single, small LSTM that is used for all parameters of the base learner (however, the authors are right that running this LSTM requires a very large cell state and input size, since the batch size of the Meta-Learner LSTM is essentially the number of parameters of the base learner). Similarly, it is stated that Andrychowicz et al. proposes to "individually [optimize] each parameter using a separate LSTM". I believe this is also not true, and that a single LSTM is used for all parameters.  In fact, ironically, NORML on the contrary appears to be using different LSTMs for each layer of the base learner (at least based on Equations 14-19, where the LSTM parameters are indexed by layer ID l, thus implying different LSTMs), unlike what is claimed in the Relate Work section ("this work uses a single meta-learner").

For me to consider increasing my rating, I would expect all points above to be addressed.

Finally, here are a few minor issues I've found:
- In Equation 16, "b_{i,\tilde{c}}" should be "b_{l,\tilde{c}}" (i.e. i should be l)
- The submission doesn't mention whether gradients are propagated into the base learner gradients (i.e. whether a first-order version of the method is used)
- 4th paragraph of the Relate Work section has a missing reference ("?")
- Table 2 is missing confidence intervals

**Experience Assessment:**

I have published in this field for several years.

**Review Assessment: Checking Correctness Of Derivations And Theory:**

N/A

**Review Assessment: Checking Correctness Of Experiments:**

I carefully checked the experiments.

**Review Assessment: Thoroughness In Paper Reading:**

I read the paper thoroughly.

---

### Official Review · AnonReviewer2 · 2019-10-20
**Official Blind Review #2**

**Rating:** 1

**Review:**

- The paper targets the scalability issue of certain meta-learning frameworks, and is therefore addressing an important and interesting topic.

- Theoretical and technical novelty is rather minimal.

- The paper writing is well beyond the ICLR level, and is honestly beyond the level required by a scientific manuscript in general. Writing needs a massive overhaul.

- There are way too many grammatical mistakes. In addition, there are citations written the wrong way -e.g. Rajeswaran (2019)-, and also the flow of the ideas has room for improvement.
- For the aforementioned reference, the author ordering is wrong, and not consistent with the way it is cited in the text.

- page 1: "while the learner can learn a set of initial task parameters are easily optimized for a new task. ": Please fix and/or clarify.

- page 2: "where the classes contained in each meta-set is disjointed".

- page 3: "The LSTM-based meta-learner proposed in this work, allow gradients to"


- "Memory-based Under review as a conference paper at ICLR 2020 methods (Ravi & Larochelle (2017)) that use all of the learner’s parameters as input to a meta-learner tend to break down when using a learner with a large number of parameters (Andrychowicz et al., 2016).": Just to clarify: The latter paper, which is criticising the former category, is older than the paper representing the former category. Is that right?

- page 2: "superior parameter updates". I see that the result has been based on classification accuracy. Notwithstanding the ablative study in the experiments secion, maybe superior parameter updates can be either replaced by a more unequivocal description of the mentioned comparison, or formally defined from there onwards.

- At the ICLR level, I do not think that all this detailed description of backpropagation would be necessary.

- Equation 1 and the description that follows: "a_l is the layer’s pre-activation output". Is a_l the post-activation output as well? Equation 1 implies so, doesn't it?

- page 3: "This limits MAML to domains where a small amount of update steps are sufficient for learning.": What do you mean by "This"? Is it to have inner loops consisting of multiple sequential updates, which is what was referred to as "preferable" in the beginning of the same paragraph? i.e. Is the MAML limitation noted in this paragraph a limitation of the vanilla MAML (plus the other versions) as well or solely due to the "preferred, yet detrimental" extension of adopting multiple sequential updates?



Minor:
- page 1: Supervised few-shot learning "aims to challenge machine learning models to ...": It does not challenge ML models; it is a specific ML paradigm.

**Experience Assessment:**

I have published one or two papers in this area.

**Review Assessment: Checking Correctness Of Derivations And Theory:**

I carefully checked the derivations and theory.

**Review Assessment: Checking Correctness Of Experiments:**

I carefully checked the experiments.

**Review Assessment: Thoroughness In Paper Reading:**

I read the paper thoroughly.

---

### Official Review · AnonReviewer3 · 2019-10-28
**Official Blind Review #3**

**Rating:** 1

**Review:**

This paper proposes a meta-learner that learns how to make parameter updates for a model on a new few-shot learning task. The proposed meta-learner is an LSTM that proposes at each time-step, a point-wise multiplier for the gradient of the hidden units and for the hidden units of the learner neural network, which are then used to compute a gradient update for the hidden-layer weights of the learner network. By not directly producing a learning rate for the gradient, the meta-learner’s parameters are only proportional to the square of the number of hidden units in the network rather than the square of the number of weights of the network. Experiments are performed on few-shot learning benchmarks. The first experiment is on Mini-ImageNet. The authors build upon the method of Sun et al, where they pre-train the network on the meta-training data and then do meta-training where the convolutional network weights are frozen and only the fully-connected layer is updated on few-shot learning tasks using their meta-learner LSTM. The other experiment is on Omniglot 20-way classification, where they consider a network with only full-connected layers and show that their meta-learner LSTM performs better than MAML.

The closest previous work to this paper is by Ravi & Larochelle, who also propose a meta-learner LSTM. The submission states about this work that “A challenge of this approach is that if you want to optimize tens of thousands of parameters, you would have a massive hidden state and input size, and will therefore require an enormous number of parameters for the meta-learner…In Andrychowicz at al. an alternative approach is introduced that avoids the aforementioned scaling problem by individually optimizing each parameter using a separate LSTM…” I don’t believe this is true.

As stated in the work by Ravi & Larochelle, “Because we want our meta-learner to produce updates for deep neural networks, which consist of tens of thousands of parameters, to prevent an explosion of meta-learner parameters we need to employ some sort of parameter sharing. Thus as in Andrychowicz et al. (2016), we share parameters across the coordinates of the learner gradient. This means each coordinate has its own hidden and cell state values but the LSTM parameters are the same across all coordinates.” Thus, Ravi & Larochelle employ something similar to Andrychowicz at al., meaning that the number of parameters in the LSTM meta-learner there is actually a constant relative to the size of the learner network.

Thus, the paper’s contribution relative to Ravi & Larochelle is to propose a LSTM meta-learner with more parameters relative to the learner model. Firstly, I think this comparison should be stated and explained clearly in the paper. Additionally, in order to prove the benefit of such an approach, I think a comparison to the work of Ravi & Larochelle with the exact experimental settings used in the submission (pre-training the convolutional network and only using the meta-learner LSTM to tune the last fully-connected layer) would be helpful in order to validate the usefulness of the extra set of parameters in their proposed meta-learner LSTM.

The submission also states that “In many cases it is preferred to have an inner loop that consists of multiple sequential updates. However the inner loop’s computational graph can become quite large if too many steps are taken. This often results in exploding and vanishing gradients since the outer loop still needs to differentiate through the entire inner loop (Aravind Rajeswaran (2019), Antoniou et al. (2018)). This limits MAML to domains where a small amount of update steps are sufficient for learning. The LSTM-based meta-learner proposed in this work, allow gradients to effectively flow through a large number of update steps. NORML can therefore be applied to a wide array of domains.” I think this statement should be validated if it is stated. Can it be shown that when making a lot of inner-loop updates, the LSTM meta-learner performs better than MAML because of overcoming the stated issues with differentiation through a long inner loop? The experiments done in the paper involve very few inner loop steps and so I don’t believe the claim is supported.

Lastly, the experimental results are not very convincing. Though the authors say they modify the method proposed in Sun et al, the results from Sun et al. are not shown in the paper. Sun et al actually seem to achieve better results than the submission with the same 1-shot and better 5-shot accuracy. Was there a reason these results are not shown in the submission? Additionally, for the Omniglot experiment, is there any reason why it was not performed with the typical convolutional network architecture? Since the original MAML results on 20-way Omniglot are with a convolutional network, using the convolutional network would make the results more meaningful relative to previous work and show that the method is more broadly applicable to convolutional networks.

I believe there are several issues with the paper as stated above. Because of these issues, it is hard to evaluate whether the idea proposed is of significant benefit.

References
Andrychowicz et al. Learning to learn by gradient descent by gradient descent. NIPS 2016.
Ravi & Larochelle. Optimization as a Model for Few-Shot Learning. ICLR 2017.
Sun et al. Meta-transfer learning for few-shot learning.

**Experience Assessment:**

I have published one or two papers in this area.

**Review Assessment: Checking Correctness Of Derivations And Theory:**

N/A

**Review Assessment: Checking Correctness Of Experiments:**

I carefully checked the experiments.

**Review Assessment: Thoroughness In Paper Reading:**

I read the paper at least twice and used my best judgement in assessing the paper.

---

### Decision · Program_Chairs · 2019-12-19

**Decision:**

Reject

**Comment:**

The paper proposes a LSTM-based meta-learning approach that learns how to update each neuron in another model for best few-shot learning performance.

The reviewers agreed that this is a worthwhile problem and the approach has merits, but that it is hard to judge the significance of the work, given limited or unclear novelty compared to the work of Ravi & Larochelle (2017) and a lack of fair baseline comparisons.

I recommend rejecting the paper for now, but encourage the authors to take the reviewers' feedback into account and submit to another venue.